# Selenium Species and Fractions in the Rock–Soil–Plant Interface of Maize (*Zea mays* L.) Grown in a Natural Ultra-Rich Se Environment

**DOI:** 10.3390/ijerph20054032

**Published:** 2023-02-24

**Authors:** Diego Armando Pinzon-Nuñez, Oliver Wiche, Zhengyu Bao, Shuyun Xie, Bolun Fan, Wenkai Zhang, Molan Tang, Huan Tian

**Affiliations:** 1Faculty of Materials Science and Chemistry, China University of Geosciences, Wuhan 430074, China; 2School of Earth Sciences, China University of Geosciences, Wuhan 430074, China; 3Ziyang Zhongdida Selenium Technology Co., Ltd., Ankang 725000, China; 4Biology/Ecology Unit, Institute of Biosciences, Technische Universität Bergakademie Freiberg, 09599 Freiberg, Germany; 5Zhejiang Institute, China University of Geosciences, Hangzhou 311305, China; 6Ankang Se-Resources Hi-Tech Co., Ltd., Ankang 725000, China; 7Scientific Research Academy of Guangxi Environment Protection, Nanning 530022, China; 8New Generation Information Technology Research Institute, Guangxi Academy of Sciences, Nanning 530007, China

**Keywords:** selenium, corn, fractions, species, HPLC-ICPMS, lower Cambrian

## Abstract

Selenium (Se) enrichments or deficiency in maize (*Zea mays* L.), one of the world’s most important staple foods and livestock feeds, can significantly affect many people’s diets, as Se is essential though harmful in excess. In particular, Se-rich maize seems to have been one of the factors that led to an outbreak of selenosis in the 1980s in Naore Valley in Ziyang County, China. Thus, this region’s geological and pedological enrichment offers some insight into the behavior of Se in naturally Se-rich crops. This study examined total Se and Se species in the grains, leaves, stalks, and roots of 11 maize plant samples, Se fractions of soils around the rhizosphere, and representative parent rock materials from Naore Valley. The results showed that total Se concentrations in the collected samples were observed in descending order of soil > leaf > root > grain > stalk. The predominant Se species detected in maize plants was SeMet. Inorganic Se forms, mainly Se(VI), decreased from root to grain, and were possibly assimilated into organic forms. Se(IV) was barely present. The natural increases of Se concentration in soils mainly affected leaf and root dry-weight biomasses of maize. In addition, Se distribution in soils markedly correlated with the weathered Se-rich bedrocks. The analyzed soils had lower Se bioavailability than rocks, with Se accumulated predominantly as recalcitrant residual Se. Thus, the maize plants grown in these natural Se-rich soils may uptake Se mainly from the oxidation and leaching of the remaining organic-sulfide-bound Se fractions. A viewpoint shift from natural Se-rich soils as menaces to possibilities for growing Se-rich agricultural products is also discussed in this study.

## 1. Introduction

Selenium (Se) is considered a micronutrient that both humans and animals need [1], but it has a very narrow range between dietary necessity and toxicity [2]. Although Se is widely dispersed in rocks, soils, sediments, waters, and plants in the environment system, and most of the Se in plants comes from the soil, in many regions, the dietary Se intake from food consumption is lower than the 40 μg day^−1^ recommended by the World Health Organization (WHO) [3].

As a trace element, the selenium content in the earth’s crust is very heterogeneous and low, with an abundance of 0.13 mg kg^−1^ [4]. Soil Se concentrations worldwide are also considerably low, mainly varying from 0.01 to 2 mg kg^−1^, with a mean value of 0.4 mg kg^−1^ [5]. Similarly, soil Se concentrations in China mostly vary from 0.02 to 3.8 mg kg^−1^, with a mean value of 0.24 mg kg^−1^ [6]. In some Se-rich areas of China, this element in soil can reach up to 87.3 mg kg^−1^ in Enshi Prefecture, Hubei Province [7], and 36.1 mg kg^−1^ in Ziyang County, Shaanxi Province [8], making these two regions the most seleniferous areas in China. However, bioavailable Se depends not only on the total Se concentration, but also on its speciation and fractions in soils. Selenium in soil mainly occurs in four states (−II, 0, IV, VI), of which inorganic Se(VI) is more accessible to plants for its high solubility and mobility, whereas Se(IV) has a higher affinity than Se(VI) for adsorption into the charged surfaces of clay minerals, thus reducing its availability for plants [9]. Under more strongly reducing conditions, selenite tends to reduce into elemental Se(0) and selenides Se(−II), which are poorly mobile and present stable and insoluble forms [10]. Different forms of organic Se in humic and fulvic acid fractions of soil organic matter might also serve as potential plant-available Se in soil [11]. In addition to the different chemical states in the soil, Se also exists in different geochemical fractions separated by multiple sequential extraction procedures (SEPs), used for extracting soil or rock Se fractions based on their differential binding capacity with various soil components [12,13]. Between these Se geochemical fractions, soluble Se and exchangeable Se fractions are commonly considered plant-available Se, but organically bound Se is also potentially available for plant uptake [14].

Maize (*Zea mays* L.), also known as corn, is the most extensively cultivated crop in the Ziyang County. Wang et al. [15] reported that the total Se concentrations in different maize tissues in Ziyang County decreased in the order of leaf > root > seed > stalk, with the highest Se concentration reaching 8.7 mg kg^−1^, 4 mg kg^−1^, 3.8 mg kg^−1^, and 2.3 mg kg^−1^, respectively. Wang et al. [16] also reported that Se concentrations of maize grain samples in Shuang’an Town, Ziyang County, were 0.9 ± 5.4 mg kg^−1^, of which 62% of the analyzed samples exceeded the Se toxicity standard (>1 mg kg^−1^), and 30% were higher than 3 mg kg^−1^. However, this is not the case for all areas in China, where Se contents in maize grains are too low to meet the dietary requirement or to present a health concern in some regions. For instance, Zhang et al. [17] recently reported that Se concentrations in the maize plant tissues in the the Wumeng Mountain area of Guizhou province were 179–232 μg kg^−1^ in roots, 23–87 μg kg^−1^ in stalks, 86–166 μg kg^−1^ in leaves, and 10–90 μg kg^−1^ in grains. In fact, Se deficiency is a widespread problem in diverse areas around the world [18]. Thus, increasing Se concentration in crops through supplementation or agronomic biofortification is necessary to supply Se dietary deficiencies [19]. It also becomes necessary to understand the transfer patterns of seleniume from rock to soil, soil to plant, and from root to grain because the accumulation of this element in edible crop parts is directly related to its concentration in soil [20]. Understanding the selenium uptake, transport, and accumulation by the crop requires an awareness of the interactions between the Se concentrations in various maize plant tissues and their total and selenium fractions in soils in the rhizosphere zone and rocks.

Aside from total Se concentrations in plants, interest in organic Se species in plants is derived from their numerous health benefits [21]. The toxicity, nutritional value, and metabolic routes of selenium in crop tissues are all dependent on the species consumed, making total concentration alone an inadequate criterion for evaluating Se-rich crop attributes [22]. Research on Se species across a variety of crops has revealed that SeMet predominates in cereal grains [23]. However, besides grains, little is known about the distribution of the Se species in different organs of maize plants grown in seleniferous soils without Se biofortification. In this study, we investigated the total Se (tSe) concentrations and Se species in different maize organs and tSe and Se fraction concentrations in their corresponding soils and parental rocks from the seleniferousarea of Naore Valley in Ziyang County, China. This research aimed to (1) elucidate the uptake, distribution, and translocation of excessive amounts of tSe and Se species in maize plants from a natural Se-rich rock–soil environment, (2) determine the geological input of Se in the soil–crop system, and (3) provide additional insights into the risk management of Se-contaminated areas.

## 2. Materials and Methods

### 2.1. Location and Geological Settings of the Study Area

Naore Village, a village of few inhabitants in the Naore valley, is located 3 km northeast of Shuang’an town, Ziyang County, Ankang City, Shaanxi Province (Figure 1). Ziyang county is geographically situated between the Qinling Mountains and the Bashan Mountains. It has a typical north subtropical humid condition, a monsoon climate, an average annual temperature of 15.1 °C, and average annual precipitation of 1066 mm.

The Lujiaping Formation of lower Cambrian strata, with particular Se geological enrichment, is partly exposed in the Ziyang area. It consists of black shales containing pyrite, intercalated stone coal, and pyritic tuff, as well as a set of light metamorphic and sedimentary shallow fine clastic rocks and clayed rocks derived from the faults system. Vanadium and phosphorous nodules are found in the siliceous limestones. The entire stratigraphic record is 700–900 m thick; however, the Se-bearing strata are only 250–400 m thick and contain an average of 20 mg kg^−1^ Se [24]. The Naore valley is aligned with a northwest-trending monoclinal structure that exposes rock strata extensively. The soil in the study area (yellow-brown soil) has a shallow depth ranging from 10 to 30 cm, was formed by the weathering of the aforementioned rocks, and has an average Se concentration of 26 mg kg^−1^ [25].

### 2.2. Collection and Preparation of Rock, Soil, and Plant Samples

The sampling took place in August 2020, during the harvest of the maize that had been planted in late April. In the upper part of Naore valley, samples of mature maize plants, the soil around roots, and related parent rock material were collected from 11 sampling points (soil–plant, rock) along a parallel transect that followed the northwest-trending monoclinal geological structure. In contrast, selenium fertilizer was once used in the lower part of Naore Valley to balance out soil’s Se concentration due to the unequal selenium concentration in the lower valley [15]. Since the soils in the lower Naore valley were no longer pristine, this study was only conducted in the upper section with no selenium fertilizer intervention. The parent rock material represents the principal types of rocks that outcrop at the surface of the research region in proximity to the soil–plant sampling locations.

The geochemical characterization of the sampling locations and the primary lithology involved the following rock types, all of which were collected from nearby exposed outcrops. These were sampled from 11 different outcrops, which are listed as follows: Sites 12, 13, 18 had weathered carbonaceous shales with occasional fine calcite lamination and high quartz contents, and similarly in site 21, there were carbonaceous slate with high amounts of pyrite and iron oxides. At sites 14 and 22, there were small layers of yellow argillaceous shale and limestone. At site 16, there were carbonaceous stone coal lenses. At sites 15 and 17, there were weathered and fresh gray carbonaceous shales interspersed with quartz masses and veins, and at site 19, there was non-pyritic gray tuffaceous slate high in carbonate contents.

According to the technical specification for soil environmental monitoring [26] and the land quality geochemical assessment [27], soil samples were collected at 0–20 cm soil depth, mainly representing the plow layer from three sub-points around the plant root, combined as one representative sample, and kept in cloth bags for transportation. At each of the 11 sampling plots, plant samples were collected as a whole by digging up the roots. The World Reference Base (WRB) for Soil Resource categorized all the soil as Luvisols, and the samples were yellow-brown tillage soils with tiny amounts of partially weathered rock debris and plant roots [28].

Samples of maize were cleaned with deionized water, sorted into their main components (root, stalk, leaves, and grains), weighed, and then oven-dried at 50 °C for 2 days. The dried plant parts were weighed again, ground to a fine powder using a plant grinder, and stored at −20 °C until analysis. After roots and detrital materials were removed, soil and rock samples were ground to pass through a 0.074 mm nylon sieve and homogenized. The pH values were measured using an FE28-pH meter at a ratio of 1:2 (rock–soil: water ratio). Contents of total organic carbon (TOC) were analyzed by the K_2_Cr_2_O_7_-H_2_SO_4_ oxidation method with 1 g of samples [29].

### 2.3. Reagents and Standards

In our analyses, reagents were used as follows: H_2_SO_4_ (96%), HNO_3_ (68%), HCl (37%), H_2_O_2_ (35%), HF (48%), NH_3_·H_2_O, methanol (HPLC quality), citric acid anhydrous (HPLC quality), and protease from Streptomyces griseus (type XIV, enzyme activity ≥ 3.5 units mg^−1^) from Sigma-Aldrich. Sodium hydroxide (NaOH), NaBH_4_, NH_3_, Tris ((hydroxymethyl) aminomethane) were purchased from Sinopharm (Beijing). Driselase from *Basidiomycetes* sp. (protein ≥ 10%) and lipase (enzyme activity ≥20 units mg^−1^) were supplied by Solarbio (Beijing). For the preparation of Se solutions, Na_2_SeO_3_ (Se(IV) 1000 mg L^−1^), Na_2_SeO_4_ (Se(VI), 1000 mg L^−1^) from Inorganic-Ventures (Christiansburg, VA, USA), Se-methionine (SeMet, >98.0%), Se-cystine (SeCys2, >98.0%), and selenomethylselenocysteine (MeSeCys, >98.0%) from TCI chemicals (Shanghai, China) were used.

### 2.4. Total Selenium (tSe) Determination in Rock, Soil, and Plant Samples

A portion of 0.5 g of dried and powdered maize sample was mixed with 2 mL concentrated HNO_3_ and 1 mL H_2_O_2_ in polytetrafluoroethylene (PTFE) vessels, allowed to react for 4 h, and then digested for 6 h at 160 °C [30]. The remaining acid solutions were reduced to Se(IV), and are listed as follows: for soil and rock digestions, a 0.1 g of sample was treated with 3 mL of concentrated mixed acid solution (HNO_3_:HF:HClO_4_, 2:0.5:0.5, *v*/*v*) in PTFE line bomb vessels and digested overnight at 180 °C. The remaining acid solutions were evaporated to almost dryness, and 2 mL of 6 mol L^−1^ HCl solution was added and heated at (<90 °C) for 1 h to reduce Se(VI) to Se(IV) before analysis of tSe. The obtained solutions were diluted with 5% HCl and stored at 4 °C until being analyzed by HG-AFS (AFS-9230, Beijing Jitian Instrument Co., Beijing, China). Reagent blanks and standard reference material GBW10045 were used for quality control.

### 2.5. Chemical Fractions of Se in Rock and Soil Samples

Five operationally defined Se fractions were determined using a modified Kulp and Pratt [13] procedure. A soil sample (1.0 g) was placed into 50 mL centrifuge tubes, and the analytical procedure was as follows: For the first extraction step (F1 = Water soluble Se), 20 mL of ultrapure water was added and shaken on a horizontal shaker at room temperature for 1.5 h. For the second step (F2 = Exchangeable Se), 10 mL of 0.1 mol L^−1^ KH_2_PO_4_-K_2_HPO_4_ buffer solution was added and the mixture was shaken for 0.5 h. The third step (F3 = Alkali-soluble Se) integrated hydrolyzable “labile” organic matter associated with humic substances and manganese, iron, and aluminum hydroxides, and was extracted through the addition of 0.1 mol L^−1^ NaOH and placed in a water bath at 90 °C for 2 h. In the fourth step, (F4 = Acid-soluble Se) the extraction of Se bound to acid-soluble labile organic matter that was not solubilized in the previous fraction is targeted, as well as Se bound to sulfides. Additionally, the residue left from F3 was oven dried (50 °C), and 0.5 g KClO_3_ and 10 mL of concentrated HCl were added and mixed for 1 h with occasional shaking. For the final step, (F5 = Residual), Se was extracted using the tSe methodology on a 0.1 g of sample from the remaining residue. Afterward, all the samples were centrifuged at 4000 rpm for 20 min, and the supernatant was filtered with 0.45 μm membrane filters. Then, 10 mL of ultrapure water was added to the tube, and centrifuging was repeated. The two supernatants were combined and stored at 4 °C until analysis. The tSe concentration in the extraction solution was measured by hydride generation-atomic fluorescence spectrometry (AFS-9230, Beijing Jitian Instrument Co., China). The extraction quality was assessed with the certified reference materials GBW07901 and GBW07400 for soil and rock, respectively. Standard and blank solutions were prepared following the same procedure together with the samples. The sequential extraction resulted in recoveries between 80 and 120%.

### 2.6. Enzyme-Soluble Selenium Species Extraction from Plant Organs

A total of 5 enzyme-soluble chemical selenium forms (selenomethionine (SeMet), selenium-cysteine (SeCys2), methylselenocysteine (MeSeCys), selenium tetravalent Se(IV), and selenium hexavalent Se(VI)) were extracted from 11 field-collected maize plants organs (grains, leaves, stalks, and roots) using enzymatic hydrolysis according to the ERM-BC210a standard material matching method [31]. As follows, 0.5 g sample was weighed in triplicates into 20 mL centrifuge tubes, and 60 mg of protease XIV, 30 mg of lipase, and 10 mL of 30 Mm Tris-HCl (pH = 7.5) were added for ultrasonic degassing. After that, they were cultured in darkness at 37 °C for 20 h, mixed well, and oscillated in a water bath oscillator at 150 r/min. The hydrolyzed samples were centrifuged at 10,000 r/min at 4 °C for 30 min. The supernatants were filtered and stored at −20 °C, and the remaining residue was added to 5 mL of hydrolysate containing 100 mg of driselase, and the above steps were repeated. The two supernatants were combined, diluted, and immediately injected into an HCLP-ICP-MS.

The concentrations of extracted Se species were determined using an ICP-MS (NexION 2000 triple quadrupole) under dynamic reaction cell mode, which was coupled to an HPLC system (Agilent 1000) equipped with an anion exchange chromatography column (Hamilton PRP-X 100). The optimized mobile phase was mixed with ultra-pure water (87%), methanol (3%), and 50 mmol L^−1^ pH = 5.5 citric acid (10%). The higher resolution mode of the ICP-QQQ allows ^80^Se to be chosen as the reported Se isotope due to its highest abundance. Methane was used as the reaction gas to avoid ^40^Ar^40^Ar^+^ ion interference.

The elution times for SeCys2, MeSeCys, Se(IV), SeMet, and Se(VI) were 2.28 min, 2.87 min, 3.29 min, 4.64 min, and 7.86 min, respectively. Unknown peaks with elution times between 6.10 and 6.37 min were seen in several columns, but their combined Se content was less than 2%; therefore, they were ignored.

The precision and accuracy obtained for tSe and Se species were assessed using ERM-BC210a for Se and SeMet analysis with enzymolysis recoveries of 85.8 ± 4.5 (Appendix A). Spike additions (0.5, 1, and 2 mg kg^−1^) also showed good agreement determination of the selenium forms with recoveries that ranged from 86% to 108%. Detection limits of the method (LOD) ranged from 0.87 μg L^−1^ to 1.70 μg L^−1^, among which the SeCys2 was the lowest (0.87 μg L^−1^), while the SeMet was the highest (1.70 μg L^−1^) (Appendix A).

### 2.7. Statistical Analysis

Relationships between tSe in, rocks, soils, maize organs, Se fractions, and Se species were identified using Pearson and Spearman correlation coefficients (r), and significance probabilities (p), linear regressions, and graphical regression analyses were performed by Origin 2017. All values presented are expressed as mean ± standard deviation in triplicates. Reported numbers are rounded for results beyond 0.1 (tSe) and 1 (Se species). Differences were considered not significant at values of *p* > 0.05.

## 3. Results and Discussion

### 3.1. Selenium Fractions in Rock–Soils Interface and Translocation Potential

The Se distribution in the different rock parent material samples and the soils derived from these strata are presented in Figure 2. Due to the high concentration of total and readily bioavailable Se fractions, the coal lenses found intercalated within the black-gray shales were identified as the most important selenium carriers, with the greatest potential to mobilize Se through soil-forming processes to maize crops. The Se concentrations in the rock strata were unevenly distributed (from 1.3 ± 0.5 to 85.5 ± 3.4 mg kg^−1^). Gray-black shales, yellow argillaceous shales, and limestones all had high tSe concentrations, while the lowest tSe was found in the gray non-pyritic tuffaceous slate (Table 1).

These results were consistent with our previous study [32], which demonstrated that in the Lower Cambrian Lujiaping Formation in Naore Village, Se was mainly associated with pyrite-bearing black shales, while the non-pyritic tuff had low Se concentrations. The soils showed a range of Se concentrations (from 7.8 ± 0.1 to 29.9 ± 2.1 mg kg^−1^), 20 to 70-fold higher than the average of 0.4 mg kg^−1^ [5], and were thus classified as ultra-high Se-rich soils in China.

The pH ranged from 6.5 to 7.9 and 6.5 to 9.3, while TOC concentrations ranged from 2.2 to 5.7% and 0.2 to 14.8% for soils and rocks, respectively (Table 1). The TOC concentrations were significantly positively correlated with the tSe in soils (r = 0.61, *p* < 0.05), but not in rocks. This suggests that biogeochemical cycling plays an important role in the examined rock–soil–plant system, where mobile-available Se forms were transferred from weathered rocks to soils, taken up by plants, and possibly returned to the soil in the form of organically bound Se with dead plant material. The biological accumulation of Se and its release into the topsoil from fallen litter upon decomposition have been shown as an important Se input in soils of typical seleniferous areas [33]. However, high TOC contents do not necessarily suggest a higher accumulation of mobile-available Se forms, as organic matter increases the organic-bound and residual fractions and conversely decreases the soluble and exchangeable Se fractions [34]. The Se encapsulated in less mobile forms can be remobilized by root exudates of [16], which may cause Se to be leached out via oxidation and organic matter to be mineralized, increasing Se mobility [35].

In the rhizosphere, microbial activity is greater than in bulk soils or weathered rocks [36], so chemical forms of Se here are governed by the release of primary and secondary metabolites from root-microbiome exudates, which alter the chemical speciation, mobility, and bioavailability of Se in the vicinity of roots. Soil Se leaching and the many alterations that roots and biological activity cause in the soil surrounding them may explain why the chemical Se fractions distribution in the parent rock materials was dissimilar to the pattern seen in soil samples.

The average mobile (bioavailable) Se content (F1 + F2) of collected rock samples was 10.8% vs. 1.8% for soils. The highest proportion of Se fractions in rocks, about 76.7%, was found in F3 + F4, which accounts for the Se bound to hydrolyzable organic matter, recalcitrant organic matter, and pyrite-sulfide minerals. In comparison, only about 21.6% of soil Se was identified in F4, whereas F5 contained the vast majority of tSe in soil (an average of 71.1%), where it was trapped as immobile residual Se in soil minerals (Appendix A). Correlations analysis between soil and rock Se fractions showed the dominant Se fractions (F4 + F5) in soils, as well as tSe, had a significant positive correlation with the dominant fractions (F3 + F4) in rocks, (r = 0.828, *p* < 0.001 and r = 0.906, *p* < 0.001, respectively) (Appendix A), which indicates the geological source of Se. The organically bound Se in Se-rich soils and weathered stone coal has been proven to be a potential source of bioavailable Se in the other selenosis area of China (Enshi) [37]. However, Se in agricultural soils may be impacted not just by bedrock weathering, but also by tillage [38] and other agricultural management practices [39].

The chemical fractions of Se in soils outside the rhizosphere zone or other inputs were not considered in this study; thus, our data did not allow further interpretation. Nevertheless, Zhu et al. [40] have already evaluated the role of human activities as the primary cause of high Se in cropland soils, and they concluded that in addition to natural factors (weathering and leaching of Se-rich rocks), the areas of higher soil Se concentrations are situated in soils heavily disturbed by human activities. The mechanical action of tillage most probably influences the movement of partially weathered rock debris and dead plant roots from the bottom to the topsoil, which in turn affects the amount of soil organic matter and accelerates the oxidation of sulfide/selenide fractions, thereby influencing tSe and labile Se forms in soils.

### 3.2. Selenium Accumulation in Maize

Since soil tSe is not always an accurate predictor of plant bioavailability, we used Pearson’s and Spearman’s correlation analyses to determine the relationship between soil tSe–Se fractions and tSe–Se species in maize organs, thereby revealing the fractions that influence most Se accumulation in maize. The results showed that there were significant positive correlations between water-soluble Se(F1) + exchangeable Se(F2) and tSe in leaves (r = 0.691 *p* < 0.05), but significant correlations with tSe in soil were not found. Alternatively, Spearman correlations with the exchangeable Se (F2) demonstrated that leaf, grain, and stalk tSe were associated with one another (r = 0.781, r = 0.709, and r = 0.681, respectively), (Appendix A). While these results are in agreement with those reported by other authors [16,41], the low concentration of these fractions, which are generally regarded as easily bioavailable for plants, suggests that maize plants could uptake Se from less bioavailable pools, even though the majority of soil Se is not readily bioavailable.

The tSe concentrations in the different plant organs were in the descending order rock > soil > leaf > root > grain > stalk (Figure 3), with the tSe concentrations in leaves being two-fold higher than roots, indicating a high efficiency of Se accumulation in the leaves and roots, as well as high Se translocation from roots into shoots. These results are consistent with the findings of Wang et al. [15], who previously investigated tSe in soil and plants from the Naore Valley. High tSe concentrations in maize leaves have also been found in Enshi [42], indicating a higher capacity for accumulating Se in the above-ground organs of maize.

As the accumulation of Se in maize plants is primarily determined by its uptake from the soil via sulfate transporters and the translocation of organic and inorganic species, the concentration of Se at the cellular level influences a variety of metabolic processes that can have an effect on plant vigor, growth, and biomass production. Therefore, the tSe concentrations in soils and plants were compared against the dry-weight accumulation in each plant organ to investigate whether Se affected the plant biomasses (Appendix A). Negative significant Pearson’s correlations were found between tSe in soils when compared with leaf and root biomasses (r = −0.672, *p* < 0.01) and (r = −0.370, *p* < 0.05), respectively, as well as with the different soil Se fractions. The root biomass had a significant inverse relationship with its tSe (r = −0.410, *p* < 0.05), and for the case of grain biomass, it was significantly and negatively correlated with its tSe (r = −0.440, *p* < 0.05)and the tSe in stalks (r = −0.346 *p* < 0.05), but significantly and positively correlated with stalk biomass (r = 0.372 *p* < 0.05) and root biomass (r = 0.501 *p* < 0.05). It is likely that high Se concentrations had a detrimental effect on the accumulation of dry matter in maize plants, affecting the organs that dealt with the highest Se accumulation, that is, roots and leaves. Toxic amounts of Se can inhibit the root system elongation in maize plants and dramatically decrease biomass accumulation in above-ground organs [43], as Se specifically reduces the ability for photosynthesis and photosynthate translocation in leaves [44], negatively impacting maize plant development [45].

### 3.3. Selenium Organ-Specific Species Distribution in Maize

A large extent of the tSe in roots, stalks, and leaves was enzyme insoluble. Figure 4 reveals that each maize organ exhibits a noticeably distinct pattern of Se species distribution. The results of the analysis of the Se species in the enzymatic hydrolysis extract of maize organs are summarized in Table 2 and listed in detail in Appendix A. The average concentrations of organic Se species (SeMet, MeSeCys, and SeCys2) were comparatively 7, 25, and 100-fold higher than the inorganic species Se(IV) + Se(VI), in rootstalks, leaves, and grains, respectively.

The enzyme-soluble Se in maize was mainly SeMet, with an average increase from 41.8% in roots to 86.7% in grains. SeMet is the Se analog of methionine, and its levels vary highly between maize genotypes [46]. Maize that thrives in seleniferous soils stores most of the Se in tissue as SeMet [47]. Inorganic species, on the other hand, were more predominant in stalks and roots at around 20%, but decreased in leaves to an average of 7.3%, while being 2.3% in grains. Inorganic Se species were found mainly as Se(VI), with very few as Se(IV).

Other organic species identified were MeSeCys, which decreased from 32.7% in roots to 2.6% in grains, and SeCys2, with the lowest average proportion found in leaves (4%) and the highest average proportion found in stalks (12%). These changes in the relative abundance of enzyme-soluble organic Se species seem to link SeCys2, and its methyl forms key building blocks for generating SeMet [48]. While MeSeCys was a major component of Se in the root, it most likely originated from biotic processes occurring in the maize crop’s below ground [49]. MeSeCys can safely be accumulated in high concentrations since it is not incorporated into proteins [50], and it is also the precursor for DMDSe (dimethyldiselenium), a key species for the plant metabolic mechanism to excrete Se excessive toxic accumulations [51].

Except for inorganic Se in grains, a positive correlation was seen between tSe and enzyme-soluble Se species in the stalks, leaves, and grains (Figure 5), suggesting that most inorganic Se forms might be transposed to organically bound Se or further reduced. Since maize is a high-stalk plant, the Se uptake exhibits a similar trend to that observed in rice, where it has been demonstrated that organic Se species, mainly SeMet, MeSeCys, and inorganic Se(VI), are rapidly loaded into the phloem and xylem vessels, yet organic species are delivered to the above-ground organs far more efficiently than Se(VI) [52]. Se(VI), if not reduced, is transported immediately to the plant’s tissues, whereas Se(IV) is uncommon because it is rapidly converted to organic forms in the maize plant’s roots [53] and translocated or rapidly reduced to Se(0) by sulfite reductase or non-enzymatically by reduced glutathione [54].

Moreover, multiple regression analyses indicated that the concentrations of Se species in maize could be predicted by tSe concentrations in any above-ground part of maize plants (i.e., stalk, leaf, grain), except for the root, where there was a general lack of correlation with tSe due to poor recoveries of enzyme-soluble Se concentrations. This indicates that uptake and conversion of Se species in roots may be distinct, and that other factors influence the extraction of enzyme-soluble Se species.

Differences in the lignin and phenol distributions among various organs of the maize plant may be responsible for the steady decline in the extracted amount of enzyme-soluble Se. Since maize roots contain more lignin than maize leaves and stalks, the lowest recoveries were observed in roots because lignin increases in concentration from leaf to root in the maize plant, an increase that is more prominent in maize than in wheat or rice [55]. In contrast, maize grains have the lowest insoluble fiber content, which explains why easily extracted Se bonds were generally higher in grains [56]. The stability of Se species may have also been impacted by reactions between Se and phenols during the extraction procedure [57], as most phenolics exist in an enzyme-insoluble state and are resistant to breakdown [58]. Future research on maize or other cereals should focus on the extraction of Se enzyme-soluble species from plant roots in order to isolate the effects of lignin and other matrix components that may result in low enzymatic extraction recoveries.

Although there were large differences in the tSe concentrations of the different maize organs as well as in the soils between sampling areas, the distribution and translocation pattern of Se species in maize plants growing in natural Se-rich soils were relatively uniform in the present study. In general, the proportion of organic species increased in SeMet, but decreased in MeSeCys from roots to grains, with varying SeCys proportions, while the proportion of inorganic Se species decreased in leaves and grains. Unlike other organic forms, SeMet is barely lost after cooking [59]. The above implies that around 560 µg of Se might be consumed from only 100 g of the analyzed maize grains, with up to 70% of that amount being accessible to the human gastrointestinal tract in the primary form of SeMet [60]. As Naore Village still has an excessively high Se intake (1801 μg Se), with 46.65% coming from cereal consumption [8], consumption of locally grown maize can dramatically raise Se ingestion above the tolerable upper intake (300 g day^−1^) [61], posing a potential health concern.

### 3.4. Current and Future Prospects of the Ultra-High Se Environment in Naore Village

In the research area, the selenosis outbreak was closely tied to maize because a prolonged drought in the 1980s led villagers to rely only on locally grown maize as their major food source, using maize stalks as fuel and their ashes as a soil fertilizer. These conditions lead to excessive Se exposure [25]. Since the years of selenosis prevalence, no new cases of human Se poisoning induced by consuming toxic Se food have been documented, as villagers’ diets have become more diverse and their exposure to toxic-Se concentrations has decreased [62]. Local consumer exposure to harmful Se concentrations from grain-based products can be further reduced by limiting direct consumption of local products and by intercropping or blending grains from locations with low and high Se concentrations.

In contrast to the infrequent reports of Se poisoning in humans, Se deficiency is becoming increasingly acknowledged as a global public health concern. Selenium dietary deficiency is either causally linked to or protects against an expanding list of diseases, such as endemic cardiovascular and enlarged joint disorders [63], infertility [64], neurodegenerative diseases [65], asthma [66], hypoglycemia [67], Chagas disease [68], and inflammatory bowel disease [69], among many others. Consequently, despite the fact that Se can be dangerous for both the environment and human health when present in excess, areas naturally enriched with Se might contribute to satisfying the global demand for Se in food. Notwithstanding some challenges, the sustainable use of Se in natural Se-rich agroecosystems can be a cost-effective and environmentally friendly approach for Se-rich agricultural resources that can provide organoselenium compounds while limiting environmental damage and decreasing selenium fertilizer use.

## 4. Conclusions

Among the five Se species examined in maize plant samples, this study determined that SeMet was the major organic species. While inorganic forms of Se diminished from root to grain, Se(VI) was identified more frequently than Se(IV). Further research is required to determine if these distributions are comparable across different maize genotypes. Additional care should be taken during the extraction and analysis of Se species in maize organs other than in grains to avoid low enzymatic extraction recoveries (as experienced in this study) due to the increases in lignin contents from leaves to roots. Most of the Se in the analyzed soils is enclosed as recalcitrant residual Se and organic-sulfide-bound Se. In contrast, Se in rocks has a comparatively higher bioavailability and is bound mainly to organic matter and sulfide minerals, with very few Se enclosed in the residual fraction. It is possible that maize plants take up a significant Se amount from the organic-sulfide-bound Se soil fraction, the weathered products derived from bedrock, or even plant litter.

In addition, excessive soil Se concentrations correlated negatively with maize plant biomass, especially roots and leaves, which accumulated more Se. Despite the low enzymatic extraction recoveries in tissues other than maize grains, this study enhances the understanding of Se species research in cereals by providing a baseline against which the results of future studies on Se fortification can be compared to natural Se-rich conditions.

## Figures and Tables

**Figure 1 ijerph-20-04032-f001:**
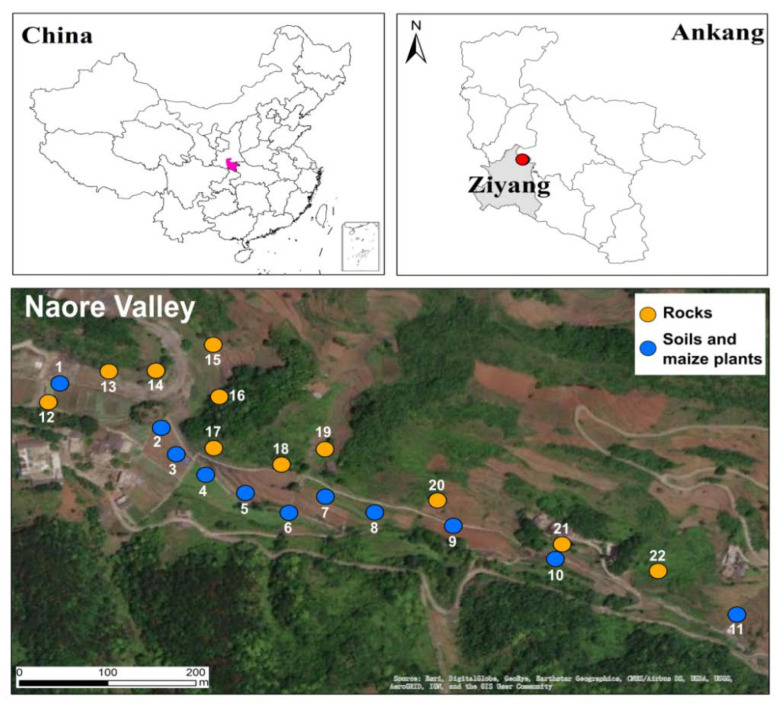
Location of the study area and sampling sites. Map of China based on GS(2019)182.

**Figure 2 ijerph-20-04032-f002:**
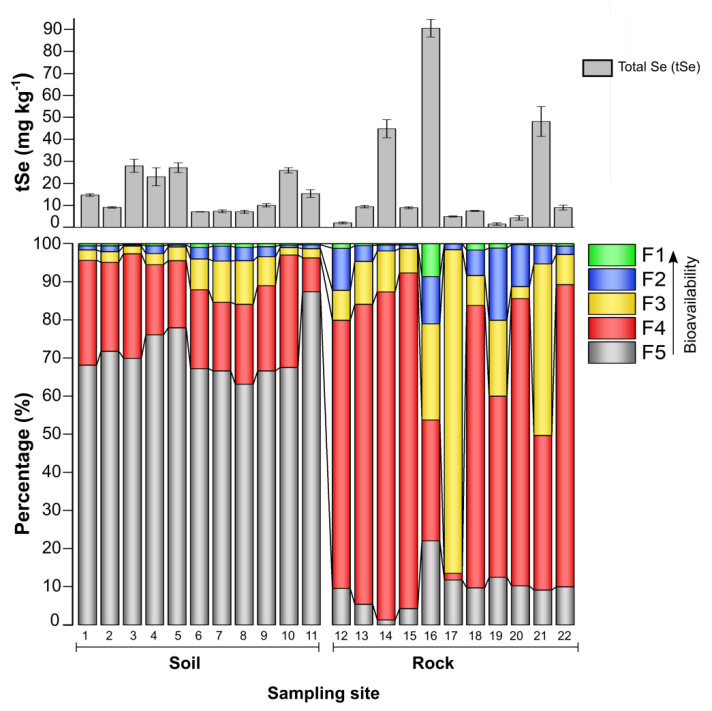
Total selenium concentrations and proportions of Se in the five operationally defined Se fractions in soil and rock samples across the study area. The bars show means ± sd (n = 3). F1: water-soluble Se, F2: Exchangeable Se, F3: Alkali-soluble Se, F4: Acid-soluble Se, and F5: Residual Se.

**Figure 3 ijerph-20-04032-f003:**
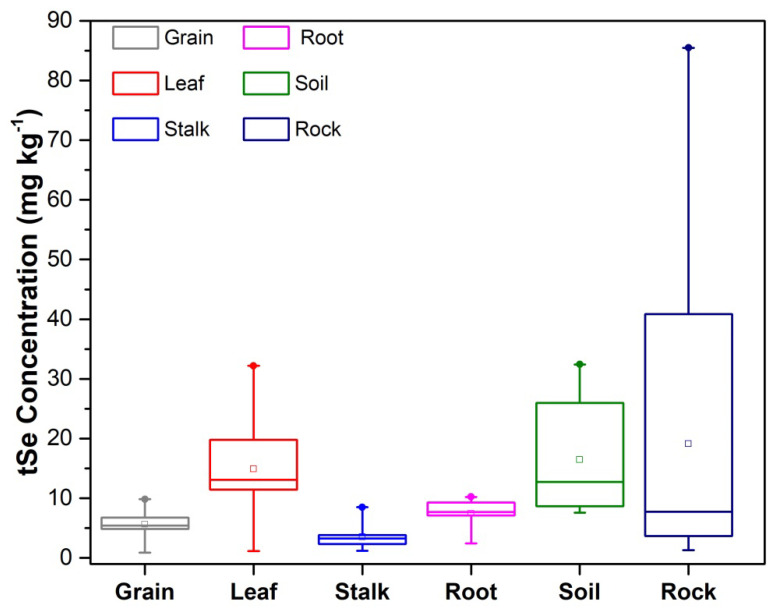
Total Se concentrations in grain, leaf, stalk, root, soil, and rocks for (n = 11) samples in triplicate. The squares are the mean values. The limits represent the minimum and maximum values, and the horizontal bottoms, middles, and top lines correspond to the 10th percentile, median, and 90th percentile values, respectively.

**Figure 4 ijerph-20-04032-f004:**
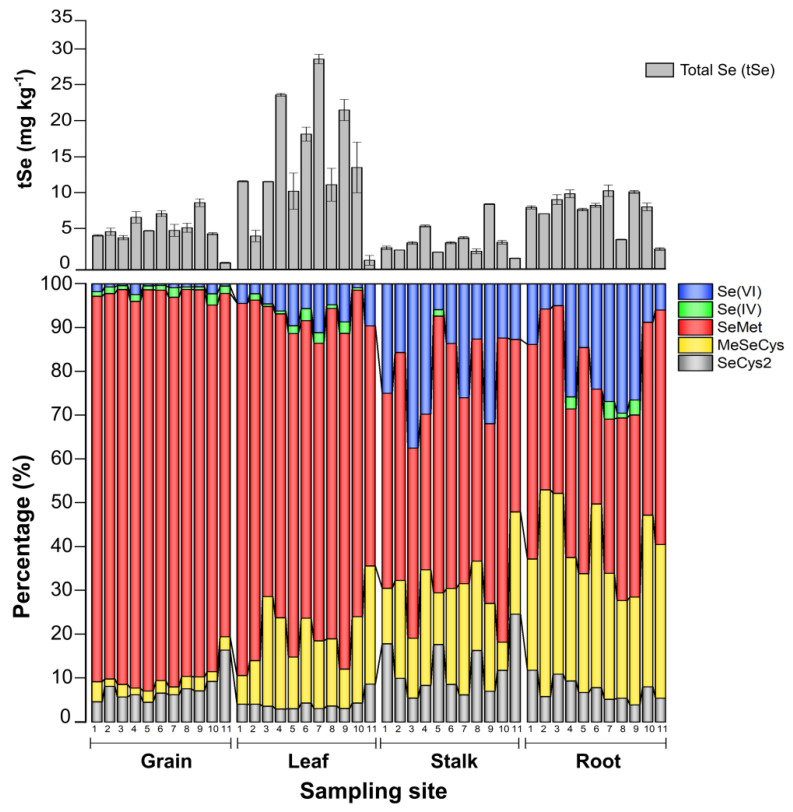
Distribution of Se enzyme-soluble species and tSe in each maize organ.

**Figure 5 ijerph-20-04032-f005:**
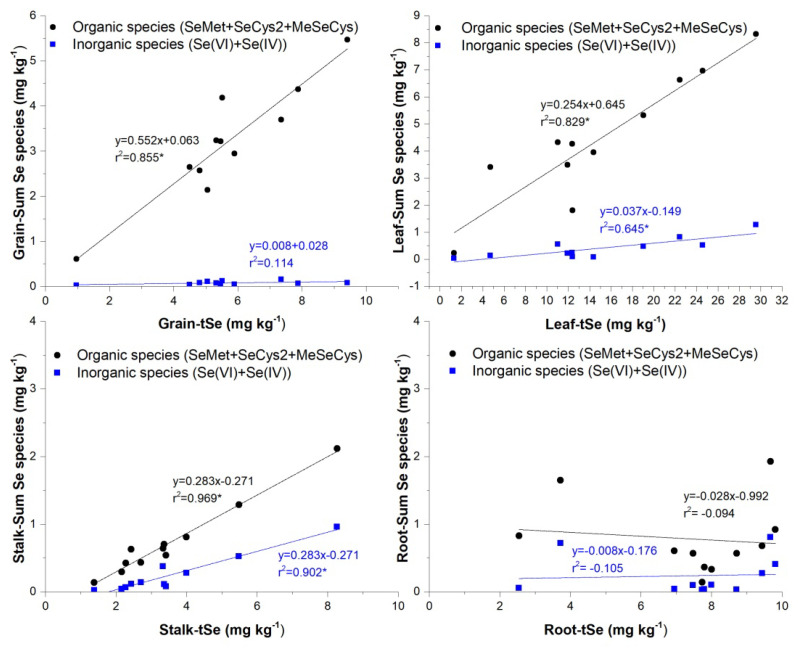
Regressions between organic and inorganic Se species when compared to tSe concentrations (* *p* < 0.05, correlation coefficient).

**Table 1 ijerph-20-04032-t001:** The concentration of tSe in rocks, soils, and pH and TOC, as well as tSe in maize organs and biomasses (dry weight per plant).

Sample	Parameter	Plot 1	Plot 2	Plot 3	Plot 4	Plot 5	Plot 6	Plot 7	Plot 8	Plot 9	Plot 10	Plot 11
Rock	No.	R12	R13	R14	R15	R16	R17	R18	R19	R20	R21	R22
	tSe (mg/kg)	2.0 ± 0.4	10.0 ± 0.5	41.5 ±4.1	7.7 ±0.5	85.5 ± 4.0	3.8 ± 1.0	6.1 ± 0.2	1.3 ± 0.3	3.7 ± 0.6	40.9 ± 6.8	7.7 ± 1.1
	pH	9.3	9.0	8.5	8.7	8.0	8.3	9.1	9.3	9.3	6.5	8.8
	TOC (%)	2.1	2.0	1.2	8.4	13.1	3.4	3.1	8.1	3.8	15.0	0.2
Soil	No.	S1	S2	S3	S4	S5	S6	S7	S8	S9	S10	S11
	tSe (mg/kg)	13.3 ± 0.6	8.4 ± 0.3	29.4 ± 3.0	22.0 ± 4.1	29.9 ± 2.2	7.8 ± 0.1	8.0 ± 0.6	8.8 ± 0.7	11.1 ± 0.8	27.8 ± 1.2	14.2 ± 1.8
	pH	7.5	7.7	7.9	8.7	7.7	7.6	7.7	7.3	6.5	7.5	7.6
	TOC (%)	4.5	3.3	4.9	5.7	7.0	2.2	5.0	3.2	5.2	4.2	5.1
Maize	No.	M1	M2	M3	M4	M5	M6	M7	M8	M9	M10	M11
	tSe-grain (mg/kg)	4.8 ± 0.1	5.3 ± 0.5	4.5 ± 0.3	7.3 ± 0.8	5.5 ± 0.05	7.9 ± 0.4	5.5 ± 0.8	5.9 ± 0.63	9.4 ± 0.5	5.0 ± 0.2	1.0 ± 0.08
	tSe-leaf (mg/kg)	12.4 ± 0.8	4.7 ± 0.05	12.4 ± 0.2	24.6 ± 2.5	11.0 ± 1.0	19.0 ± 0.7	29.6 ± 2.3	11.9 ± 1.5	22.4 ± 3.5	14.3 ± 0.7	1.3 ± 0.2
	tSe-stalk (mg/kg)	2.7 ± 0.2	2.4 ± 0.2	3.3 ± 0.1	5.5 ± 0.03	2.2 ± 0.13	3.4 ± 0.1	4.0 ± 0.3	2.3 ± 0.06	8.3 ± 0.2	3.4 ± 0.04	1.4 ± 0.2
	tSe-root (mg/kg)	7.7 ± 0.6	6.9 ± 0.3	8.7 ± 0.5	9.4 ± 0.2	7.5 ± 0.2	8.0 ± 0.7	9.8 ± 0.05	3.7 ± 0.2	9.7 ± 0.5	7.8 ± 0.2	2.5 ± 0.1
	Biomass-grain (g)	414.8	385.1	282.1	298.2	101.6	322.2	369.9	356.3	515.8	188.7	415.0
	Biomass-leaf (g)	94.9	106.3	88.9	63.2	100.1	146.3	128.7	125.8	145.1	84.5	143.6
	Biomass-stalk (g)	94.5	93.5	136.6	121.5	98.3	106.5	124.0	116.2	114.4	145.8	133.5
	Biomass-root (g)	63.0	109.62	34.2	33.8	39.3	99.5	35.8	49.0	56.2	85.5	96.2

**Table 2 ijerph-20-04032-t002:** Se species in enzymatic hydrolysis extracts and tSe of maize samples (in Se mg kg^−1^ and % Se of species for tSe content).

Maize Organ	tSe	SUM-Se Extracted	Organic Species						Inorganic Species		
			SeMet		SeCys2		MeSeCys		Se(IV) + Se(VI)		Extraction Efficiency
	(mg/kg)	(mg/kg)	(mg/kg)	(%)	(mg/kg)	(%)	(mg/kg)	(%)	(mg/kg)	(%)	(%)
Grain	5.6 ± 2.2	5.6 ± 2.3	5.0 ± 2.1	86.7 ± 3.5	0.43 ± 0.17	7.4 ± 3.3	0.13 ± 0.1	2.62 ± 0.86	0.06 ± 0.03	2.3 ± 1.3	98.7 ± 9.8
Leaf	14.9 ± 8.4	8.0 ± 4.6	6.2 ± 3.5	80.0 ± 8.3	0.33 ± 0.15	4.0 ± 1.6	1.2 ± 0.82	16.1 ± 6.5	0.3 ± 0.28	7.3 ± 3.8	54.1 ± 15.5
Stalk	3.5 ± 1.9	1.5 ± 1.2	0.81 ± 0.57	48.1 ± 10.4	0.17 ± 0.1	12.0 ± 6.0	0.3 ± 0.31	18.3 ± 6.3	0.19 ± 0.21	20.1 ± 10.0	37.5 ± 10.6
Root	7.4 ± 2.3	1.5 ± 1.1	0.76 ± 0.58	41.8 ± 7.9	0.11 ± 0.07	7.3 ± 2.5	0.46 ± 0.29	32.7 ± 8.4	0.18 ± 0.21	17.9 ± 10.7	25.4 ± 25.6

## Data Availability

The datasets utilized in this investigation are accessible upon request from the corresponding authors.

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
