# Peer review of "Selenium Species and Fractions in the Rock–Soil–Plant Interface of Maize (Zea mays L.) Grown in a Natural Ultra-Rich Se Environment"

_ijerph, 2023, doi:10.3390/ijerph20054032_

Round 1

Reviewer 1 Report

IJERPH-Distribution of selenium fractions and species in the rock-soil-plant interface of maize from Naore Valley selenosis region

The present study focused on a selenosis region as Naore Valley in Shanxi, China, and collected 11 maize plants and 11 nearby parent rocks to figure out the Se distributions of Se fractions in rocks and soils and of Se speciation in plant tissues. Overall, the written English is difficult to understand and follow, and the highlights on study site (a selenosis region) and the rock-soil-plant interface were not clearly presented in the current version. Moreover, the present manuscript was more descriptive, which fell in local interest rather than a broad interest to readers. Thus, I will recommend to re-write those data to explore more comprehensive explanations there.

Minor concerns:

1-    Title: “Zea mays” should be italic, which should be checked through out the manuscript; “Naore Valley Selenosis Region” is confused to me, please rewrite this title to make it clear.

2-    L17: Please check the maize Latin name with missing a “L.”

3-    L18: “selenium-enriches” should be “selenium-rich” since the Se was naturally existed in soil here. Please also check this through out the manuscript.

4-    L20-21: Confused here on the sample collection.

5-    L29: “Se inorganic forms” should be “Inorganic Se forms”.

6-    L135: How to define the corresponding parent rock material here?

7-    L152: How to define and collect the rhizosphere soil? According to the present description, I assumed the present soil samples as bulk soils instead of rhizosphere soil. Please confirm this.

8-    L161: It is not clear to refer “as describe below” here.

9-    L245-248: Since Se fertilizers were applied in the present study site, how to distinguish the Se source from weathered rocks or Se fertilizer?

10- L262-264: Not clear. Please explain how to cause the different patterns between rock and soil here.

11- Figure 2: Generally the residue (F5) was predominant in rocks, which is totally different with the present study. Please explain why.

12- Table 1: How to determine pH in rock? Also, please put rock samples before soil sample, which will be consistent with the rock-soil-plant logic.

13- Table 2: Why to have big differences on extraction efficiency among different tissues?

14- Figure 5: There should have deepen explanations among the differences in plant tissues. Moreover, in root, the sum of organic Se and inorganic Se was significantly lower than that of total Se. So what is the gap? Where to go on those Se, as unidentified Se?

15- L423-436: No data or references to support?

Author Response

Hello, thank you very much for your precious time spent on reviewing our manuscript, we really appreciate it.

Regarding the English of the manuscript, we did our best to improve it throughout the entire document; as a result, you will notice that many sentences and paragraphs have been rewritten in a more straightforward manner. However, we apologize in advance if you still find errors in the use of the English language. We also rework the abstract, some portions of the results, and the conclusions in order to broaden the work's appeal to readers and its relevance to other researchers. Thank you for highlighting this point. The manuscript has undergone many changes so we also send the clean version to facilitate the reading.

Now considering the review's main point:

  1. Regarding the title, we made a change to increase attention.
  2. Many thanks for clarifying rizhosphere soils concept, for avoiding any confusion regarding rizhosphere soils, the word was used more cautiously, and it was clarified in the methodology that soils surrounding the root (or around the rhizosphere zone) were collected.
  3. The large variations in extraction efficiencies among maize tissues were described in further detail as changes in insoluble fiber concentration, specifically lignin contents, which is greatest in roots. This was supported by literature.
  4. Furthermore, the differences between Se fractions in soils also were further clarified.
  5. The sample location information was further rewritten and clarified
  6. we also review the references in the last part, added new and write it as a new section
  7. The order of the table was also changed it
  8. Accordingly also to the other reviewer we have added a further section for reagents.

The remaining minor corrections were accordingly assesed.

Thank you for all of your recommendations; we hope the new version had improved. Have a wonderful weekend ~

Reviewer 2 Report

This is the review of “Distribution of Selenium Fractions and Species in the Rock- soil-plant Interface of Maize (Zea mays L.) from Naore Valley Selenosis Region”. General: in this work, total Se, Se fractions, and Se species distributions in maize plant samples, including grains, leaves, stalks, roots, rhizosphere soils, and the most representative parent rock materials from Naore Valley, Zi-yang County, China were investigated. This work provided lots of useful data for Se pollution and human health, and takes an important role in understanding the Se pollution. However, there is few Se pollution assessment in paper. Thus, I think this manuscript can be accepted for publication in this journal unless addressed the following comments.     

Comments and questions:

1.       Abstract: I think the abstract is very normal and cannot highlight the special points of this paper. So, I suggest authors should rewrite abstract.

2.       The introduction should be modified to keep it more logical, and please itemize the contributions of your manuscript at the end of the introduction. In addition, most of the references cited in introduction were published before 2020, please add more newest citation.

3.       Methods: the regents purity information should be provided in methods, and the full name of the test method should be given when it first appears.

4.       Results and discussion: Line 236-238, this sentence should be rewritten to make it easier to understand.

5.       Results and discussion: Fig.4, the extracted Se should be part of the total Se, so I suggest authors revise Fig.4.  

6.       Results and discussion: Line 360-361, this sentence should be modified to make it clearer.

7.       Results and Discussion: I suggest authors revise this section to highlight the contribution of this article, instead of great lengths to describe literature. In addition, what is the difference between this article and other relevant review articles?

8.       Results and Discussion: I suggest that more Se pollution assessment of studied area should be given.

9.       Conclusion: I suggest author rewrite the conclusion to make this work get better summary.

10.    Last, please check the English grammar, and modify it.

Author Response

Dear reviewer, Hello, thank you so much for taking the time to read the work, identify its weak parts, and offer all of your recommendations, trust, and support. Thanks !

In general;

As you can see, the manuscript has been extensively revised, particularly in terms of English style and grammar, although we apologize if you still detect errors in the English usage, and all your suggestions have been implemented throughly.
As the soils are naturally enriched with Se, we attempt to avoid mentioning pollution in the manuscript. Instead, the reader may find a full discussion on the dual toxicity-benefit of selenium in the environment and the sustainable utilization of Se-rich agroecosystems. Since Se was primarly considered a menace but recently the shift in paradigm is changing towards the multiple benefits of Se so there is an ongoing demand for Se global foodstuff. This perspective opens up the debate of agricultural goods with naturally or intentionally increased Se content, in this case maize, to a wider audience.

In specific,

About the different comments and questions ;

  1. The abstract has been rewritten in a more accessible and interesting way. Also results are presented more clear.
  2. The English style of many sentences of the introduction was modified, and a couple of 2020-2022new references were added.
  3. A new section ‘Reagents and Standards’ was added to the manuscript in which it the regents purity information appears as well as the iformation of the supplier.
  4. Following your suggestion, Figure 4 was reviewed.
  5. Results and discussion section was throughly proofread and reviewed, many sentences and paragraphs were rewritten to showcast the results for a more wider audience.
  6. While the primary purpose of our research was descriptive, we also took this occasion to show that maize was a contributing cause to selenosis issues in the 1980s and to highlight the ways in which the situation has changed since then. This section was separated of the main results narrative, and added as ‘Current and future prospects of the ultra-high Se environment in Naore village’ .
  7. Following also your recommendation, the conclusion were completely re-written to show in a better and more summarized way the results of the present study.

Other reviews of the English in specific sentences were done accordingly.

Thank you very much and hope the revised version of the manuscript had improved and be of your liking. We surely worked hard to improve it. Have a wonderful weekend  and thank you very much~

Round 2

Reviewer 1 Report

Thanks the authors to make a great improvement on the manuscript and almost addressed my all concerns. Now it could be accepted after fixing the following issues:

1-L28: "Se inorganic Se forms" should be "Inorganic Se from";

2-L55: No keywords in the Abstract section;

3-L57: What is "animals humans"?

4-L143-144: Apparently, the present statement as "without Se biofortification" is un-consistent with the information in the previous version as Se fertilizer used in the study site. Please explain why?

Author Response

Attached is the cover letter for Editors and reviewers

Reviewer 2 Report

  • Authors have  revised this manuscript and responsed my comments, and can be published in this journal.

Author Response

Dear Reviewer#2,

We sincerely appreciate the time you spent reviewing the manuscript and the trust you placed in it, as well as the effort you put into the review process.  The manuscript underwent further proofreading, and minor mistakes have been assessed. Thanks a lot.